# Area-level deprivation and individual-level socioeconomic correlates of the diabetes care cascade among black south africans in uMgungundlovu, KwaZulu-Natal, South Africa

Sanele Listen Mandlenkosi Madela[1], Nigel Walsh Harriman[2]*, Ronel Sewpaul[3], Anthony David Mbewu[4], David R Williams[2,5], Sibusiso Sifunda[3], Thabang Manyaapelo[6], Anam Nyembezi[7], Sasiragha Priscilla Reddy[8]

1 Expectra Health Solutions, Dundee, South Africa, 2 Social and Behavioral Sciences Department, Harvard T.H. Chan School of Public Health, Boston, Massachusetts, United States of America, 3 Human and Social Capabilities Division, Human Sciences Research Council, Cape Town, South Africa, 4 Department of Internal Medicine, Sefako Makgatho Health Sciences University, Ga-Rankuwa, South Africa, 5 Department of African and American Studies, Harvard University, Cambridge, Massachusetts, United States of America, 6 Africa Health Research Institute, Durban, South Africa, 7 School of Public Health, University of the Western Cape, Cape Town, South Africa, 8 College of Health Sciences, University of KwaZulu-Natal, South Africa

* nharriman@hsph.harvard.edu

## Abstract

South Africa is experiencing a rapidly growing diabetes epidemic that threatens its health-care system. Research on the determinants of diabetes in South Africa receives considerable attention due to the lifestyle changes accompanying South Africa's rapid urbanization since the fall of Apartheid. However, few studies have investigated how segments of the Black South African population, who continue to endure Apartheid's institutional discriminatory legacy, experience this transition. This paper explores the association between individual and area-level socioeconomic status and diabetes prevalence, awareness, treatment, and control within a sample of Black South Africans aged 45 years or older in three municipalities in KwaZulu-Natal. Cross-sectional data were collected on 3,685 participants from February 2017 to February 2018. Individual-level socioeconomic status was assessed with employment status and educational attainment. Area-level deprivation was measured using the most recent South African Multidimensional Poverty Index scores. Covariates included age, sex, BMI, and hypertension diagnosis. The prevalence of diabetes was 23% (n = 830). Of those, 769 were aware of their diagnosis, 629 were receiving treatment, and 404 had their diabetes controlled. Compared to those with no formal education, Black South Africans with some high school education had increased diabetes prevalence, and those who had completed high school had lower prevalence of treatment receipt. Employment status was negatively associated with diabetes prevalence. Black South Africans living in more deprived wards had lower diabetes prevalence, and those residing in wards that became more deprived from 2001 to 2011 had a higher prevalence diabetes, as well as diabetic control. Results from this study can assist policymakers and practitioners in identifying modifiable risk factors for diabetes among Black South Africans to intervene on. Potential community-based interventions include those focused on patient empowerment and

**Data Availability Statement:** All relevant data are within the paper and its Supporting Information files.

**Funding:** SLM's work in the HealthRise Program was funded by the Medtronic Foundation (https://foundation.medtronic.com/us-en/index/global-health.html) and Abt Associates Incorporated (https://www.abtassociates.com) Grant Agreement No 47169. The funders had no role in study design, data collection and analysis, decision to publish, or preparation of the manuscript.

**Competing interests:** The authors have declared that no competing interests exist.

linkages to care. Such interventions should act in concert with policy changes, such as expanding the existing sugar-sweetened beverage tax.

## Introduction

South Africa is in the midst of a rapidly growing diabetes epidemic that poses a major threat to its public healthcare system. According to the International Diabetes Foundation, the number of people living with diabetes in South Africa in 2021 has doubled since 2011 and quadrupled since 2000 [1]. In 2019, the age-adjusted prevalence of diabetes in South Africa was 12.7%, up from 5.5% in 2010; and the highest of any sub-Saharan African nation measured [2–4]. These trends are expected to continue in South Africa and other developing countries. One study by Shaw et al projected a 69% increase in the number of adults with diabetes in developing countries from 2010 to 2030 [5]. Other analyses have suggested that the prevalence of diabetes in Africa will more than double by 2040 [6]. In South Africa, the number of those living with diabetes is expected to nearly double again by 2045 [1,7]. The mortality due to diabetes in South Africa is also striking. From 2016 until it was last measured by Statistics South Africa in 2018, diabetes was the second leading natural cause of death in South Africa, accounting for between 5.5% and 5.9% of all deaths [8]. South Africa's rising diabetes prevalence and mortality will also have an increasingly severe impact on its healthcare system. According to an analysis by Erzse and colleagues, the economic burden of diabetes in 2018 was 21.8 billion ZAR, which is projected to increase to 35.1 billion ZAR in 2030 [9]. This burden is particularly urgent given the South African Government's implementation of National Health Insurance (NHI); a funding scheme aimed at providing universal health coverage. The economic impact of diabetes is a major threat to this nascent and fragile healthcare financing system, which has already faced a plethora of barriers to implementation [10,11].

The increased prevalence of diabetes and other non-communicable diseases (NCDs) is inextricably tied to South Africa's epidemiological transition since the democratic dispensation in 1994. Omran's theory of the Epidemiological Transition posits that as societies become more developed, the burden of disease shifts from communicable disease to non-communicable disease [12]. In South Africa, it seems that different segments of the population experience the epidemiologic transition at varying rates, based on various socio-ecological factors, including their demographics, and changes in geography and socioeconomic status that accompany economic development [13]. This may partly relate to gross inequities in South Africa, whose most recent Gini coefficient of 0.7 placed it as the most unequal country in the world [14]. Indeed, nationally representative data from South Africa from various years have consistently indicated that diabetes is patterned along a socioeconomic gradient aligned with the epidemiological transition, whereby those with higher individual-level markers of socioeconomic status (i.e. income, household wealth, education), report a higher prevalence of disease [15,16]. While these and other studies have investigated the socioeconomic gradient associated with diabetes at a population level, few studies have attempted to identify the determinants within subgroups of the population, an analysis that is congruent with the propositions of Omran's theory of the epidemiological transition. One such population is Black South Africans, who have endured and continue to contend with the legacy of Apartheid. The current economic implications of the structural racism engineered by Apartheid-era policies are stark, with vast racial inequities present along various individual-level markers of socioeconomic status [17,18].

These socioeconomic disparities are also heavily tied to health outcomes and can help elucidate some of the drivers of the extant racial disparities in diabetes outcomes observed at the national level [19]. According to nationally representative data, Black South Africans had the highest prevalence of undiagnosed diabetes, a key metric of the diabetes care cascade, of any racial group in South Africa [20]. A systematic review of the literature on the prevalence of diabetes in South Africa revealed marked variation in the prevalence of diabetes not only across racial groups but within them as well. This within-race variation was particularly pronounced within Black South Africans and was further graded by geographic setting. Interestingly, the presence of urban-rural disparities was not present at the aggregate level or for any other racial group, a finding hypothesized by the authors to reflect that Black South Africans were at a different stage in the epidemiological transition and that different patterns of the transition were occurring within the group as well [21,22].

Changes in area-level socioeconomic status are also a key marker of the epidemiological transition and have consistently demonstrated significant associations with diabetes across a variety of settings. Findings from the United States, Germany, the United Kingdom, and Sweden, developed countries which are further along in the epidemiological transition, have indicated that area-level deprivation is a risk factor for diabetes [23–26]. The association between diabetes and area deprivation in these countries has been hypothesized to be related to reduced access to health-promoting institutions and preventive care facilities that are present in deprived areas [27]. Conversely, in developing countries, the opposite pattern is typically observed. A longstanding hypothesis for this socioeconomic gradient has been that increases in a sedentary lifestyle and access to unhealthy food, and behaviors, such as alcohol intake and smoking, are all risk factors for non-communicable diseases, that accompany urbanization [28]. Changes in area-level socioeconomic status are particularly relevant to the Black South African population group, who faced mass racial segregation through forced removals to city outskirts and rural areas under the Group Areas Act [29]. After the fall of Apartheid, the subsequent urbanization and aforementioned behavioral exposures experienced by this group as a result of this segregation may place them at greater risk for developing diabetes. Indeed, existing research has indicated the age-standardized prevalence of diabetes among Black urban-dwelling South Africans in Cape Town had risen by 53% between 1990 and 2009 [30]. Despite this, few published analyses have attempted to explore the association between area-level markers of socioeconomic status and its association with chronic disease within the Black South African sub-population.

Although scientific literature on the prevalence and social determinants of diabetes in South Africa is abundant, there is a paucity of studies that have focused on these same determinants within the Black African subpopulation. An empirical examination of the socioeconomic variation within this population would provide public health policymakers and practitioners with valuable information to develop targeted initiatives to improve the health of Black South Africans. Toward that end, the current study aims to assess the relative contributions of individual and area-level socioeconomic status to diabetes prevalence, awareness, treatment, and control within a sample of 3,685 Black South Africans living in the KwaZulu-Natal province.

## Materials and methods

### Data

This paper analyzes secondary data from the HealthRise South Africa Study, a community-based program sponsored by the Medtronic Foundation designed to improve screening, diagnosis, management, and control of hypertension and diabetes in underserved communities [31]. Cross-sectional data were collected from February 2017 to February 2018 by Expectra

Health Solutions, a non-governmental organization accredited by the Health & Welfare Sector Education Training Authority in community health work. Participants lived in three subdistricts of the uMgungundlovu district in KwaZulu-Natal province: the Msunduzi, uMshwathi, and Mkhambathini. These subdistricts were deliberately selected following consultation with the Institute for Health Metrics and Evaluation (IHME) and the KwaZulu-Natal Department of Health. Eligible catchment areas included those with minimal or no existing Department of Health CHWs. The selected areas were also required to include residents living in rural informal, rural formal (farm), urban formal, and urban informal areas. The final selected areas had a catchment population of 163,995. Further details of the selection criteria are described elsewhere [32–35]. To ensure compliant and valid data collection from participants, twenty-five community health workers were trained to follow the KwaZulu-Natal Department of Health Adult Primary Care (APC) guidelines [36,37]. Participant data were collected at the following three locations: 10,658 door-to-door household visits (n = 7,954), twenty-four community outreach campaigns (n = 2,353), and sixteen workplace visits (n = 525).

**Dependent variables.** This analysis had four dichotomous outcomes: diabetes prevalence, awareness, treatment, and control. Random blood glucose was measured following the KwaZulu-Natal Department of Health Adult Primary Care (APC) guidelines. All participants aged 45 and older had their blood glucose measured. A detailed description of the screening eligibility criteria for those under 45 is described elsewhere and is not a focus of these analyses [32].

Following the International Diabetes Foundation guidelines, the presence of diabetes was defined as either: 1) a random blood glucose measurement of 11.1 mmol/L at the time of the study or 2) if the participant reported a healthcare professional had previously diagnosed them with diabetes [38]. Awareness was conditional upon having diabetes; participants were defined as aware of their diabetes if they reported that a healthcare professional had previously diagnosed them. Among those who were aware, those who reported they were currently taking medication to treat their diabetes were considered in treatment. Control of one's diabetes was then conditional upon receiving treatment and defined as having a random blood glucose measurement below 10.0 mmol/L at the time of the study.

**Independent variables.** We measured participant age as a continuous variable and sex as a dichotomous variable. Educational attainment was divided into the following categories: no formal education, some primary school, incomplete high school, completed high school, and tertiary. Employment status was dichotomous (unemployed vs. employed). We used the following cut points to describe participant Body Mass Index: under 18.5 kg/m$^2$ (underweight), between 18.5 kg/m$^2$ and 24.9 kg/m$^2$ (normal weight), between 25 kg/m$^2$ and 29.9 kg/m$^2$ (overweight), and 30 kg/m$^2$ or higher (obese). We also included a variable for the presence of hypertension, defined as either 1) having a measured blood pressure at the time of data collection of 140/90 mmHg and higher or 2) reporting that a healthcare professional had previously diagnosed the individual with hypertension.

**South African multidimensional poverty index.** Data was collected on the municipality and ward where participants resided. This information was then linked to the 2011 (most recent available) and 2001 census multidimensional poverty data to assess how deprived the ward was in which a participant resided. We employed the South African Multidimensional Poverty Index (SAMPI), a metric developed by Statistics South Africa to measure area-level deprivation unique to the South African context [39]. SAMPI scores range from 0 to 1 with higher scores indicating stronger area-level deprivation. The SAMPI is comprised of four household indicators of deprivation–health, education, economic activity, and standard of living–the specific cut points and weights of these indicators are described elsewhere [39]. The SAMPI score for a geographic area is the product of two metrics: 1) the poverty headcount–the proportion of households in the area that are deprived on at least one of the four indicators,

and 2) the poverty intensity–the average proportion of the four indicators that deprived households are deprived in. For example, if 25% of the households in an area were deprived, and the deprived households were deprived on three of the four indicators (75%), on average, the SAMPI score for this area would be 0.25*0.75 = 0.1875.

The SAMPI has been used to predict various dimensions of health in South Africa, including self-rated health and cardiovascular disease [40,41]. In this analysis, we utilized both absolute and relative measures of area area-level deprivation. We operationalized 2011 SAMPI scores as a continuous measure to reflect absolute area-level deprivation. In our analyses, we utilized log-transformed scores to ensure our variable was linear to our outcome. Regarding relative deprivation, SAMPI scores were modeled as an ordinal variable with cut points based on the quintiles of the distribution of SAMPI scores for all South Africa. None of the participants in the study lived in wards in the 1st (least deprived) and 5th (most deprived) quintiles of the 2011 SAMPI scores. Our final relative 2011 SAMPI variable was dichotomous, comparing the 2nd and 3rd quintiles (combined due to low cell counts) to the 4th (more deprived) quintile.

Our assessments of the change in area-level deprivation also reflect a ward's absolute and relative change. We computed SAMPI quintile variables for 2001 and 2011 and then constructed a dichotomous variable to determine if a ward's relative SAMPI quintile had worsened or remained the same from 2001 to 2011 (no wards had relative improvement). Our measure of absolute change reflected the difference in SAMPI score from 2001 to 2011 (SAMPI 2011 score–SAMPI 2001 score) and multiplied by 100 for enhanced interpretability; this allowed us to assess the association between our outcomes and a 0.01-unit difference in change in SAMPI score.

## Statistical analyses

The analyses presented in this paper were restricted to Black South Africans who were 45 years and older and whose data was obtained during door-to-door survey visits (n = 3,685). This restricted sample was chosen due to 1) only participants aged 45 and older had their random blood glucose measured, 2) there were low counts of other racial groups, and 3) area-level data for a participant's residence could only be collected during the door-to-door surveying process.

First, we produced descriptive statistics for our dependent and independent variables including any missing data on our independent variables. We then imputed these missing values using multiple imputation with chained equations (MICE). We produced and performed analyses on 10 imputations of our dataset.

Following the imputation procedure, we conducted a series of ten hierarchical multilevel Poisson regression models with robust standard errors and randomly varying intercepts for ward to estimate prevalence ratios for each of our four diabetes outcomes (Prevalence, Awareness, Treatment, and Control) using Stata SE. Model 1 estimated the association between the diabetes outcome of interest and demographic variables (age and sex). Model 2 added individual-level proxies of SES (education level and employment status). Model 3 added an area-level deprivation measure to model 2. Finally, in model 4, individual risk factors (BMI and hypertension prevalence) were added to the parameters in model 3. For each diabetes outcome, models 3 and 4 were each run with "a," "b," "c," and "d" counterparts for our different measures of absolute and relative area-level deprivation. The "a" models utilized the 2011 SAMPI quintile, "b" models reflect continuous log-transformed 2011 SAMPI score, "c" models reflect the relative change in SAMPI quintile from 2001 to 2011, and "d" models reflect the absolute change in SAMPI score from 2001 to 2011.

**Table 1. Sample description.**

| Variable | N | Percentage |
|---|---|---|
| **Diabetes Prevalence** | | |
| No previous or current diabetes | 2855 | 77% |
| Currently or previously diagnosed with diabetes | 830 | 23% |
| **Diabetes Awareness** | | |
| Unaware | 61 | 7% |
| Aware | 769 | 93% |
| **Diabetes Treatment** | | |
| Untreated | 140 | 18% |
| Treated | 629 | 82% |
| **Diabetes Control** | | |
| Uncontrolled | 225 | 36% |
| Controlled | 404 | 64% |
| **Sex** | | |
| Male | 905 | 24.6% |
| Female | 2780 | 75.4% |
| Missing | 0 | 0.0% |
| **Education Level** | | |
| None | 1543 | 41.9% |
| Some Primary School | 1088 | 29.5% |
| High School Incomplete | 510 | 13.8% |
| High School Complete | 313 | 8.5% |
| Tertiary | 79 | 2.1% |
| Missing | 152 | 4.1% |
| **Employment Status** | | |
| Unemployed | 2871 | 77.9% |
| Employed | 628 | 17.0% |
| Missing | 186 | 5.0% |
| **BMI** | | |
| Underweight <18.5 kg/m2 | 375 | 10.2% |
| Normal weight 18.5–24.9 kg/m2 | 1210 | 32.8% |
| Overweight 25–29.9 kg/m2 | 829 | 22.5% |
| Obese > = 30 kg/m2 | 1047 | 28.4% |
| Missing | 224 | 6.1% |
| **Hypertension** | | |
| No Hypertension | 1316 | 35.7% |
| Current or Previous Hypertension | 2369 | 64.3% |
| Missing | 0 | 0.0% |
| **2011 SAMPI Quintile** | | |
| Q2 (Less Deprived) | 59 | 1.6% |
| Q3 | 860 | 23.3% |
| Q4 (More Deprived) | 2369 | 64.3% |
| Missing | 397 | 10.8% |
| **SAMPI Quintile Change (2001 to 2011)** | | |
| No change | 2852 | 77.4% |
| Quintile Worsened | 436 | 11.8% |
| Missing | 397 | 10.8% |
| | **Mean (SD)** | **Median** |

*(Continued)*

**Table 1.** (Continued)

| Variable | N | Percentage |
|---|---|---|
| **Age** | 60.46 (11.19) | 59 |
| **SAMPI 2011 Score** | 0.04 (0.01) | 0.04 |
| **SAMPI Absolute Change (2001 to 2011)** | -0.05 (0.02) | -0.07 |

## Ethics statement

All individuals consented to participate in the study. The study was conducted in accordance with the International Ethical Practice for Research with Human Subjects. Written informed consent was obtained from all participants and/or their legally authorized representatives. Participants who were unable to read and/or write had the consent forms read out to them and a fingerprint was obtained. Ethical approval for the study was obtained from the Human Sciences Research Council of South Africa (HSRC) Research Ethics Committee (REC)—Protocol Number: REC 4/21/09/16. The HSRC REC is registered with the South African National Health Research Ethics Council (REC-290808–015). The HSRC REC has US Office for Human Research Protections (OHRP) Federal-wide Assurance (FWA Organization No. 0000 6347).

## Results

### Sample description

In Table 1, we present the descriptive statistics of the sample. The mean age of the sample was 60.46 (SD 11.19) and the majority were female (75.4%). The most frequent responses for educational attainment were as follows: no formal education (41.9%), some primary school (29.5%), high school incomplete (13.8%), high school completed (8.5%), and tertiary education (2.1%). Over three-quarters of the sample was unemployed (77.9%). One-third (32.8%) of the sample was categorized as having a normal weight (BMI: 18.5–24.9 kg/m$^2$), the next highest percentage being those who were obese ($> = 30$ kg/m$^2$) at 28.4%. Twenty-two percent of the sample was overweight (25–29.9 kg/m$^2$), and 10.2% of the sample was underweight ($<18.5$ kg/m$^2$). The majority of the sample was either screened as hypertensive at the time of the examination or had been previously diagnosed with hypertension (64.3%). Most respondents resided in wards in the 4$^{th}$ quintile of South Africa's 2011 SAMPI score distribution (64.3%). Approximately 2% of the sample lived in wards in the 2$^{nd}$ quintile, and 23.3% lived in wards in the 3$^{rd}$ quintile. The mean 2011 SAMPI score for our sample was 0.042 (SD 0.01). None of our respondents lived in wards whose SAMPI quintile improved from 2001 to 2011, but 11.8% lived in wards whose SAMPI quintile worsened from 2001 to 2011. Most individuals lived in wards whose SAMPI scores improved from 2001 to 2001; the mean change score was -0.054 (SD 0.02) The prevalence (current or previous diagnosis) of diabetes in our sample was 23% (n = 830). Of those who were diabetic, 93% had been previously diagnosed (n = 769). Among those aware (previously diagnosed), 82% were currently receiving treatment (n = 629). Of those in treatment, 64% had their diabetes controlled (n = 404).

### Diabetes prevalence

Table 2 presents the results of the multi-level Poisson regression models with robust standard errors for diabetes prevalence (n = 3,685) among Black South Africans aged 45 and above. Across all models, we observed that diabetes prevalence was positively associated with age.

In model 1, females had a higher prevalence of diabetes than males (PR 1.294, p = 0.012), and this significant association persisted until adjusting for individual-level risk factors in

**Table 2. Prevalence Ratios (PR) and Standard Errors (SE) for diabetes prevalence (n = 3,685).**

| Diabetes Prevalence | Model 1 - Age + Sex PR | SE | Model 2 - Model 1 + SES PR | SE | Model 3a - Model 2 + SAMPI Quintile PR | SE | Model 4a - Model 3a + BMI & HTN Prevalence PR | SE | Model 3b - Model 2 + 2011 SAMPI Score PR | SE | Model 4b - Model 3a + BMI & HTN Prevalence PR | SE | Model 3c - Model 2 + SAMPI Quintile Change PR | SE | Model 4c - Model 3b + BMI & HTN Prevalence PR | SE | Model 3d - Model 2 + SAMPI Change PR | SE | Model 4d - Model 3c + BMI & HTN Prevalence PR | SE |
|---|---|---|---|---|---|---|---|---|---|---|---|---|---|---|---|---|---|---|---|---|
| VARIABLES | | | | | | | | | | | | | | | | | | | | |
| Age | 1.020*** | 0.003 | 1.019*** | 0.003 | 1.019*** | 0.003 | 1.010** | 0.003 | 1.019*** | 0.003 | 1.010** | 0.003 | 1.019*** | 0.003 | 1.010** | 0.004 | 1.019*** | 0.003 | 1.010** | 0.004 |
| Male | ref | | ref | | ref | | ref | | ref | | ref | | ref | | ref | | ref | | Ref | |
| Female | 1.294* | 0.134 | 1.280* | 0.137 | 1.276* | 0.138 | 1.138 | 0.093 | 1.272* | 0.137 | 1.127 | 0.091 | 1.277* | 0.139 | 1.143 | 0.097 | 1.276* | 0.139 | 1.142 | 0.097 |
| No education | | | ref | | ref | | ref | | ref | | ref | | ref | | ref | | ref | | ref | |
| Some primary education | | | 1.066 | 0.081 | 1.066 | 0.075 | 1.097 | 0.078 | 1.062 | 0.075 | 1.096 | 0.078 | 1.075 | 0.081 | 1.105 | 0.084 | 1.071 | 0.078 | 1.103 | 0.080 |
| High school incomplete | | | 1.139 | 0.082 | 1.144 | 0.088 | 1.233*** | 0.074 | 1.145 | 0.091 | 1.238*** | 0.072 | 1.149 | 0.089 | 1.238*** | 0.074 | 1.146 | 0.089 | 1.233*** | 0.076 |
| High school complete | | | 0.964 | 0.153 | 0.976 | 0.159 | 1.055 | 0.150 | 0.969 | 0.160 | 1.046 | 0.146 | 0.967 | 0.160 | 1.043 | 0.154 | 0.965 | 0.160 | 1.040 | 0.155 |
| Tertiary | | | 1.269 | 0.325 | 1.280 | 0.325 | 1.252 | 0.281 | 1.283 | 0.325 | 1.263 | 0.278 | 1.266 | 0.318 | 1.236 | 0.276 | 1.269 | 0.317 | 1.232 | 0.277 |
| Unemployed | | | ref | | ref | | ref | | ref | | ref | | ref | | ref | | ref | | ref | |
| Employed | | | 0.775* | 0.092 | 0.776* | 0.096 | 0.817 | 0.087 | 0.777* | 0.097 | 0.816 | 0.086 | 0.770* | 0.092 | 0.809* | 0.080 | 0.771* | 0.093 | 0.809* | 0.081 |
| Underweight <18.5 kg/m2 | | | | | | | | | | | | | | | | | | | | |
| Normal weight 18.5–24.9 kg/m2 | | | | | | | 1.004 | 0.069 | | | 1.001 | 0.068 | | | 1.002 | 0.069 | | | 1.001 | 0.068 |
| Overweight 25–29.9 kg/m2 | | | | | | | 1.124 | 0.121 | | | 1.123 | 0.119 | | | 1.128 | 0.122 | | | 1.127 | 0.122 |
| Obese > = 30 kg/m2 | | | | | | | 1.263* | 0.149 | | | 1.265* | 0.151 | | | 1.262 | 0.151 | | | 1.264 | 0.154 |
| No HTN | | | | | | | ref | | | | ref | | | | ref | | | | ref | |
| Current or Previous HTN | | | | | | | 4.179*** | 0.650 | | | 4.182*** | 0.660 | | | 4.161*** | 0.643 | | | 4.169*** | 0.643 |
| Q2 and Q3 | | | | | ref | | ref | | | | | | | | | | | | | |
| Q4 (More deprived) | | | | | 0.836 | 0.082 | 0.840** | 0.054 | | | | | | | | | | | | |
| ln(2011 SAMPI Score) | | | | | | | | | 0.707*** | 0.063 | 0.681*** | 0.070 | | | | | | | | |
| No Change in Quintile | | | | | | | | | | | | | ref | | ref | | | | | |
| Quintile Worsened | | | | | | | | | | | | | 1.183 | 0.135 | 1.158 | 0.098 | | | | |
| SAMPI Absolute Change from 2001 to 2011 | | | | | | | | | | | | | | | | | 1.024 | 0.016 | 1.024* | 0.012 |

* p < 0.05.

** p < 0.01.

*** p < 0.001.

models 4a, b, c, and d. In model 2, which adjusts for participant demographics and socioeconomic status, employment status was significantly associated with diabetes prevalence. Specifically, employed individuals had a 22% reduced prevalence of diabetes compared to those who were unemployed (PR 0.775, p = 0.032). The significant negative association between employment and diabetes prevalence persisted until adjusting for individual-level risk factors in models 4a and 4b, however, in our models where we assessed changes in area-level deprivation (4c and 4d), the employment differences in diabetes prevalence remained significant.

While relative area-level deprivation was not associated with diabetes prevalence in model 3a, after adjusting for individual-level risk factors for diabetes in model 4a, we observed significant differences in diabetes prevalence across wards. In this model, individuals who lived in wards with higher relative area-level deprivation had lower diabetes prevalence than those in wards with lower deprivation (Q4 vs. Q2 and Q3: PR 0.84, p = 0.006). Consistent with our measure of relative deprivation in 2011, absolute deprivation was also significantly negatively associated with the prevalence of diabetes in our sample (PR 0.681, p<0.001). In this model, Black South Africans aged 45 and older who had some level of high school completed had 1.23 times the prevalence of diabetes compared to those who had no education (PR 1.233, p<0.001). Across all remaining models, we observed that significant differences in diabetes prevalence by educational attainment only emerged in model 4, after adjusting for both area-level measures of deprivation and individual-level risk factors for diabetes.

We observed that our measure of relative change in area-level deprivation was not significantly associated with diabetes prevalence in models 3c and 4c. Absolute change in area-level deprivation displayed a divergent pattern. In the fully-adjusted model 4d, a 0.01 unit increase in SAMPI score from 2001 to 2011 was associated with a 2.4% increase in the prevalence of diabetes (PR 1.024, p = 0.041).

### Diabetes awareness

In Table 3, we present the multi-level Poisson regression models for diabetes awareness, which restricts the sample to those who have diabetes (n = 830). Across all models, we did not observe any significant associations between awareness of one's diabetes and age, sex, or educational attainment.

Employment status was protective of receiving a previous diabetes diagnosis (PR 0.964, p = 0.018) and this association persisted until adjusting for individual-level risk factors for diabetes in models 4a, b, c, and d.

In model 3a, residents in wards with higher relative area-level deprivation had a lower prevalence of a previous diabetes diagnosis than those in wards with lower deprivation (Q4 vs. Q2 and Q3: PR 0.951, p = 0.039). However, this association was marginally significant in the fully adjusted model 4a. Absolute deprivation in 2011 was significantly negatively associated with a previous diabetes diagnosis in both model 3b (PR 0.908, p = 0.007) and 4b (PR 0.93, p = 0.021). We did not observe significant differences in the prevalence of a previous diabetes diagnosis by our measures of change in area-level deprivation.

### Diabetes treatment

The results of our regression models for the treatment of diabetes are presented in Table 4. Here again, our analytic sample is restricted to 769, as treatment of diabetes is conditional upon being aware. In model 1, and all subsequent models, we observed that females had a higher prevalence of receiving treatment for their diabetes compared to males. In these models, we did not observe that receipt of diabetes treatment was associated with age, employment, or area-level deprivation.

**Table 3. Prevalence Ratios (PR) and Standard Errors (SE) for diabetes awareness (n = 830).**

| Diabetes Awareness | Model 1 - Age + Sex | | Model 2 - Model 1 + SES | | Model 3a - Model 2 + SAMPI Quintile | | Model 4a - Model 3a + BMI & HTN Prevalence | | Model 3b - Model 2 + 2011 SAMPI Score | | Model 4b - Model 3a + BMI & HTN Prevalence | | Model 3c - Model 2 + SAMPI Quintile Change | | Model 4c - Model 3b + BMI & HTN Prevalence | | Model 3d - Model 2 + SAMPI Change | | Model 4d - Model 3c + BMI & HTN Prevalence | |
|---|---|---|---|---|---|---|---|---|---|---|---|---|---|---|---|---|---|---|---|---|
| VARIABLES | PR | SE | PR | SE | PR | SE | PR | SE | PR | SE | PR | SE | PR | SE | PR | SE | PR | SE | PR | SE |
| Age | 1.001 | 0.001 | 1.001 | 0.001 | 1.001 | 0.001 | 1.000 | 0.001 | 1.001 | 0.001 | 1.000 | 0.001 | 1.001 | 0.001 | 1.000 | 0.001 | 1.001 | 0.001 | 1.000 | 0.001 |
| Male | ref | | ref | | ref | | ref | | ref | | ref | | ref | | ref | | ref | | ref | |
| Female | 1.007 | 0.014 | 1.010 | 0.012 | 1.005 | 0.012 | 1.003 | 0.013 | 1.003 | 0.012 | 1.003 | 0.013 | 1.007 | 0.012 | 1.006 | 0.012 | 1.006 | 0.012 | 1.005 | 0.012 |
| No education | | | ref | | ref | | ref | | ref | | ref | | ref | | ref | | ref | | ref | |
| Some primary education | | | 0.996 | 0.037 | 0.996 | 0.037 | 0.991 | 0.033 | 0.998 | 0.037 | 0.993 | 0.033 | 0.999 | 0.039 | 0.992 | 0.034 | 0.999 | 0.039 | 0.992 | 0.034 |
| High school incomplete | | | 1.022 | 0.027 | 1.022 | 0.026 | 1.019 | 0.022 | 1.023 | 0.027 | 1.020 | 0.022 | 1.023 | 0.028 | 1.020 | 0.023 | 1.023 | 0.028 | 1.020 | 0.023 |
| High school complete | | | 1.049 | 0.044 | 1.052 | 0.045 | 1.043 | 0.037 | 1.050 | 0.044 | 1.043 | 0.037 | 1.046 | 0.046 | 1.039 | 0.038 | 1.047 | 0.045 | 1.040 | 0.037 |
| Tertiary | | | 0.914 | 0.091 | 0.913 | 0.087 | 0.914 | 0.075 | 0.919 | 0.085 | 0.919 | 0.074 | 0.910 | 0.088 | 0.912 | 0.076 | 0.911 | 0.087 | 0.913 | 0.075 |
| Unemployed | | | ref | | ref | | ref | | ref | | ref | | ref | | ref | | ref | | ref | |
| Employed | | | 0.964* | 0.015 | 0.965* | 0.014 | 0.980 | 0.014 | 0.966* | 0.015 | 0.980 | 0.014 | 0.964* | 0.014 | 0.980 | 0.014 | 0.964* | 0.014 | 0.979 | 0.014 |
| Underweight <18.5 kg/m2 | | | | | | | ref | | | | ref | | | | ref | | | | ref | |
| Normal weight 18.5–24.9 kg/m2 | | | | | | | 0.973 | 0.024 | | | 0.973 | 0.024 | | | 0.974 | 0.024 | | | 0.973 | 0.024 |
| Overweight 25–29.9 kg/m2 | | | | | | | 0.959* | 0.017 | | | 0.958* | 0.017 | | | 0.960* | 0.018 | | | 0.959* | 0.017 |
| Obese > = 30 kg/m2 | | | | | | | 0.940** | 0.021 | | | 0.940** | 0.021 | | | 0.938** | 0.022 | | | 0.938** | 0.022 |
| No HTN | | | | | | | ref | | | | ref | | | | ref | | | | ref | |
| Current or Previous HTN | | | | | | | 1.267*** | 0.061 | | | 1.264*** | 0.061 | | | 1.270*** | 0.062 | | | 1.269** | 0.062 |
| Q2 and Q3 | | | | | ref | | ref | | | | | | | | | | | | | |
| Q4 (More deprived) | | | | | 0.951* | 0.023 | 0.964 | 0.019 | | | | | | | | | | | | |
| ln(2011 SAMPI Score) | | | | | | | | | 0.908** | 0.033 | 0.930* | 0.029 | | | | | | | | |
| No Change in Quintile | | | | | | | | | | | | | ref | | ref | | | | | |
| Quintile Worsened | | | | | | | | | | | | | 1.046 | 0.031 | 1.030 | 0.027 | | | | |
| SAMPI Absolute Change from 2001 to 2011 | | | | | | | | | | | | | | | | | 1.008 | 0.004 | 1.005 | 0.003 |

* p < 0.05.

** p < 0.01.

*** p < 0.001.

**Table 4. Prevalence Ratios (PR) and Standard Errors (SE) for diabetes treatment (n = 769).**

| Diabetes Treatment | Model 1 - Age + Sex | | Model 2 - Model 1 + SES | | Model 3a - Model 2 + SAMPI Quintile | | Model 4a - Model 3a + BMI & HTN Prevalence | | Model 3b - Model 2 + 2011 SAMPI Score | | Model 4b - Model 3a + BMI & HTN Prevalence | | Model 3c - Model 2 + SAMPI Quintile Change | | Model 4c - Model 3b + BMI & HTN Prevalence | | Model 3d - Model 2 + SAMPI Change | | Model 4d - Model 3c + BMI & HTN Prevalence | |
|---|---|---|---|---|---|---|---|---|---|---|---|---|---|---|---|---|---|---|---|---|
| VARIABLES | PR | SE | PR | SE | PR | SE | PR | SE | PR | SE | PR | SE | PR | SE | PR | SE | PR | SE | PR | SE |
| Age | 1.005 | 0.003 | 1.005 | 0.003 | 1.005 | 0.003 | 1.005 | 0.003 | 1.004 | 0.003 | 1.005 | 0.003 | 1.004 | 0.003 | 1.005 | 0.003 | 1.004 | 0.003 | 1.005 | 0.003 |
| Male | ref | | ref | | ref | | ref | | ref | | ref | | ref | | ref | | ref | | ref | |
| Female | 1.100** | 0.035 | 1.089*** | 0.032 | 1.094** | 0.033 | 1.069* | 0.031 | 1.095*** | 0.033 | 1.070* | 0.030 | 1.096** | 0.033 | 1.072* | 0.031 | 1.093*** | 0.032 | 1.069* | 0.030 |
| No education | | | ref | | ref | | ref | | ref | | ref | | ref | | ref | | ref | | ref | |
| Some primary education | | | 0.956 | 0.055 | 0.956 | 0.052 | 0.976 | 0.055 | 0.954 | 0.051 | 0.974 | 0.054 | 0.947 | 0.050 | 0.967 | 0.052 | 0.952 | 0.051 | 0.973 | 0.054 |
| High school incomplete | | | 0.950 | 0.061 | 0.950 | 0.060 | 0.964 | 0.062 | 0.950 | 0.059 | 0.963 | 0.061 | 0.946 | 0.058 | 0.960 | 0.060 | 0.949 | 0.059 | 0.963 | 0.061 |
| High school complete | | | 0.841** | 0.056 | 0.839** | 0.056 | 0.851** | 0.052 | 0.840* | 0.058 | 0.852* | 0.054 | 0.847* | 0.062 | 0.860* | 0.058 | 0.842* | 0.059 | 0.854* | 0.056 |
| Tertiary | | | 1.080 | 0.146 | 1.085 | 0.150 | 1.101 | 0.173 | 1.080 | 0.146 | 1.097 | 0.169 | 1.102 | 0.155 | 1.119 | 0.180 | 1.090 | 0.155 | 1.107 | 0.179 |
| Unemployed | | | ref | | ref | | ref | | ref | | ref | | ref | | ref | | ref | | ref | |
| Employed | | | 0.971 | 0.080 | 0.971 | 0.079 | 0.968 | 0.077 | 0.970 | 0.079 | 0.967 | 0.077 | 0.972 | 0.077 | 0.968 | 0.075 | 0.972 | 0.079 | 0.968 | 0.077 |
| Underweight <18.5 kg/m2 | | | | | | | ref | | | | ref | | | | ref | | | | ref | |
| Normal weight 18.5–24.9 kg/m2 | | | | | | | 1.077 | 0.056 | | | 1.078 | 0.056 | | | 1.078 | 0.055 | | | 1.078 | 0.057 |
| Overweight 25–29.9 kg/m2 | | | | | | | 1.045 | 0.070 | | | 1.046 | 0.070 | | | 1.042 | 0.069 | | | 1.045 | 0.070 |
| Obese >= 30 kg/m2 | | | | | | | 1.205** | 0.070 | | | 1.204** | 0.070 | | | 1.207*** | 0.067 | | | 1.207** | 0.070 |
| No HTN | | | | | | | ref | | | | ref | | | | ref | | | | ref | |
| Current or Previous HTN | | | | | | | 1.014 | 0.077 | | | 1.016 | 0.077 | | | 1.017 | 0.076 | | | 1.014 | 0.075 |
| Q2 and Q3 | | | | | ref | | ref | | | | | | | | | | | | | |
| Q4 (More deprived) | | | | | 1.045 | 0.071 | 1.033 | 0.073 | | | | | | | | | | | | |
| ln(2011 SAMPI Score) | | | | | | | | | 1.096 | 0.138 | 1.076 | 0.134 | | | | | | | | |
| No Change in Quintile | | | | | | | | | | | | | ref | | ref | | | | | |
| Quintile Worsened | | | | | | | | | | | | | 0.878 | 0.089 | 0.878 | 0.086 | | | | |
| SAMPI Absolute Change from 2001 to 2011 | | | | | | | | | | | | | | | | | 0.990 | 0.010 | 0.991 | 0.010 |

* p < 0.05.

** p < 0.01.

*** p < 0.001.

In model 2, diabetic adults who had completed high school had a 16% reduced prevalence of receiving treatment for their diabetes, compared to those with no formal education (PR 0.841, p = 0.01), and this association persisted in all subsequent models, with little change to its effect size or significance.

### Diabetes control

In Table 5, we analyzed the prevalence of diabetic control, conditional upon receiving treatment (n = 629). Across all models, we did not observe any significant associations between diabetes control and age, employment status, and educational attainment.

While our most up-to-date measures of area-level deprivation were not associated with control of one's diabetes, results for our measures of relative and absolute change in area-level deprivation showed a clearer pattern. In our fully adjusted model 4c, diabetic Black South Africans living in wards that had become more deprived since 2001 relative to other South African wards had a 20% higher prevalence of diabetes control compared to those living in wards that had not (PR 1.196, p = 0.005). Moreover, in model 4d, a 0.01 unit increase in SAMPI score from 2001 to 2011 was associated with a 3.8% increase in the prevalence of diabetes control compared to those living in wards that had not (PR 1.038, p<0.001).

## Discussion

Our study sought to examine the individual and area-level socioeconomic gradients in diabetes prevalence, awareness, treatment, and control among a sample of Black South Africans, aged 45 and older, living in the uMgungundlovu district of the KwaZulu-Natal province. Ours is one of few existing studies that investigate the association between educational attainment and employment status with the diabetes care cascade outcomes among Black South Africans. Furthermore, to our knowledge, this study is unique in its application of the South African Multi-Dimensional Poverty Index to assess both cross-sectional and historical changes in area-level deprivation.

The relationship between individual measures of socioeconomic status and diabetes was complex and deserves discussion. Broadly, our results revealed variable directions of diabetes' socioeconomic gradient regarding employment and educational attainment. We found that, in our fully adjusted models including change in deprivation over time, employed South Africans had lower prevalence of diabetes than their unemployed counterparts. This finding is conceptually aligned with the social determinants of health framework, whereby individuals with higher socioeconomic status have increased access to healthy foods, health education, health insurance, and health-promoting institutions. Interestingly, existing research conducted in other LMICs has ofttimes reported the opposite pattern–that higher socioeconomic status individuals have higher rates of diabetes [28]. While the other associations reported in this analysis are generally consistent with the existing literature on this topic, it may be that given our sample is exclusively of Black South Africans, we are capturing a racialized experience (for example, high levels of labor market exclusion) that might not be observable in other studies that do not stratify by race. It is possible that the Black South Africans in our sample who have diabetes may be less likely to find employment. Of the limited extant literature on the topic, one study in South Africa using nationally representative data from the 2018 General Household Survey found that having diabetes was associated with worse labor market outcomes among aging populations [42]. It is plausible that older Black South Africans, who already face barriers to employment due to their race and age, face additional labor market exclusion when diagnosed with diabetes, which may require workplace accommodations, such as time off from work that employers are unwilling to provide. This is especially relevant given the

**Table 5. Prevalence Ratios (PR) and Standard Errors (SE) for diabetes control (n = 629).**

| Diabetes Control | Model 1 - Age + Sex | | Model 2 - Model 1 + SES | | Model 3a - Model 2 + SAMPI Quintile | | Model 4a - Model 3a + BMI & HTN Prevalence | | Model 3b - Model 2 + 2011 SAMPI Score | | Model 4b - Model 3a + BMI & HTN Prevalence | | Model 3c - Model 2 + SAMPI Quintile Change | | Model 4c - Model 3b + BMI & HTN Prevalence | | Model 3d - Model 2 + SAMPI Change | | Model 4d - Model 3c + BMI & HTN Prevalence | |
|---|---|---|---|---|---|---|---|---|---|---|---|---|---|---|---|---|---|---|---|---|
| VARIABLES | PR | SE | PR | SE | PR | SE | PR | SE | PR | SE | PR | SE | PR | SE | PR | SE | PR | SE | PR | SE |
| Age | 1.004 | 0.003 | 1.003 | 0.003 | 1.003 | 0.003 | 1.001 | 0.002 | 1.003 | 0.002 | 1.002 | 0.002 | 1.003 | 0.003 | 1.002 | 0.002 | 1.004 | 0.003 | 1.002 | 0.002 |
| Male | ref | | ref | | ref | | ref | | ref | | ref | | ref | | ref | | ref | | ref | |
| Female | 0.905 | 0.050 | 0.896* | 0.047 | 0.891* | 0.052 | 0.914 | 0.051 | 0.881* | 0.053 | 0.904 | 0.052 | 0.885* | 0.051 | 0.907 | 0.050 | 0.879* | 0.053 | 0.901 | 0.052 |
| No education | ref | | ref | | ref | | ref | | ref | | ref | | ref | | ref | | ref | | ref | |
| Some primary education | | | 1.030 | 0.044 | 1.029 | 0.044 | 0.998 | 0.046 | 1.033 | 0.047 | 1.003 | 0.049 | 1.036 | 0.044 | 1.003 | 0.044 | 1.036 | 0.046 | 1.004 | 0.046 |
| High school incomplete | | | 0.867 | 0.075 | 0.866 | 0.074 | 0.853 | 0.072 | 0.866 | 0.080 | 0.853 | 0.078 | 0.867 | 0.076 | 0.851 | 0.074 | 0.862 | 0.079 | 0.848 | 0.076 |
| High school complete | | | 1.104 | 0.117 | 1.106 | 0.117 | 1.086 | 0.128 | 1.107 | 0.117 | 1.090 | 0.130 | 1.086 | 0.105 | 1.066 | 0.115 | 1.088 | 0.101 | 1.071 | 0.113 |
| Tertiary | | | 0.848 | 0.239 | 0.841 | 0.233 | 0.817 | 0.219 | 0.843 | 0.228 | 0.819 | 0.214 | 0.812 | 0.208 | 0.782 | 0.190 | 0.800 | 0.209 | 0.772 | 0.191 |
| Unemployed | | | ref | | ref | | ref | | ref | | ref | | ref | | ref | | ref | | ref | |
| Employed | | | 0.926 | 0.103 | 0.928 | 0.103 | 0.938 | 0.105 | 0.932 | 0.103 | 0.942 | 0.106 | 0.932 | 0.102 | 0.944 | 0.102 | 0.932 | 0.102 | 0.944 | 0.103 |
| Underweight <18.5 kg/m2 | | | | | | | ref | | | | ref | | | | ref | | | | ref | |
| Normal weight 18.5–24.9 kg/m2 | | | | | | | 0.848* | 0.056 | | | 0.842* | 0.057 | | | 0.847* | 0.057 | | | 0.842* | 0.057 |
| Overweight 25–29.9 kg/m2 | | | | | | | 0.810* | 0.086 | | | 0.803* | 0.086 | | | 0.816 | 0.085 | | | 0.811* | 0.084 |
| Obese >= 30 kg/m2 | | | | | | | 0.715*** | 0.057 | | | 0.715*** | 0.054 | | | 0.710*** | 0.058 | | | 0.711*** | 0.055 |
| No HTN | | | | | | | ref | | | | ref | | | | ref | | | | ref | |
| Current or Previous HTN | | | | | | | 1.124* | 0.067 | | | 1.113 | 0.067 | | | 1.115 | 0.072 | | | 1.108 | 0.067 |
| Q2 and Q3 | | | | | ref | | ref | | | | | | | | | | | | | |
| Q4 (More deprived) | | | | | 0.951 | 0.110 | 0.963 | 0.115 | | | | | | | | | | | | |
| ln(2011 SAMPI Score) | | | | | | | | | 0.808 | 0.091 | 0.819 | 0.092 | | | | | | | | |
| No Change in Quintile | | | | | | | | | | | | | ref | | ref | | | | | |
| Quintile Worsened | | | | | | | | | | | | | 1.192** | 0.074 | 1.196** | 0.077 | | | | |
| SAMPI Absolute Change from 2001 to 2011 | | | | | | | | | | | | | | | | | 1.038*** | 0.012 | 1.038*** | 0.011 |

\* p < 0.05.

\*\* p < 0.01.

\*\*\* p < 0.001.

occupational segregation experienced by Black South Africans, whose limited employment opportunities are restricted to low-paying jobs which require manual labor [43]. The physical demand of this work is likely extremely difficult for older individuals who have diabetes, which increases in prevalence with age; and potential employers may neglect this population as a result. On the other hand, given the sample's age, our measure of employment may have also captured those who were unemployed because they were retired. It is plausible that in this case, employment status was capturing residual effects of age on diabetes prevalence. Given these findings, future research in Black South African communities should aim to longitudinally investigate the relationship between diabetes and employment to establish the direction of this association.

Our results regarding the relationship between education and diabetes prevalence are also noteworthy. We observed that Black South Africans who had completed some high school education had higher diabetes prevalence than those without any formal education. Interestingly, this significant association only emerged once we adjusted for traditional risk factors for diabetes, including hypertension and body mass index. How risk factors for diabetes impact the onset of the disease are complex and do not operate linearly. While the positive association between education and diabetes has been observed in other LMICs, interestingly, we observed that Black South Africans with tertiary education did not significantly differ from those with no formal education [44]. While higher socioeconomic status increases the likelihood of consuming unhealthy foods and living a sedentary lifestyle (risk factors for diabetes), individuals of higher socioeconomic status also have increased access to health education and health-promoting institutions [45]. It may be that Black South Africans who have attained some level of high school education are exposed to the risks that accompany increased socioeconomic status in terms of diet and sedentary behavior, but still face barriers to healthcare access and utilization which might buffer their adverse effects. Lending credence to this explanation were our findings that diabetic Black South Africans who had completed high school had a lower prevalence of treatment receipt for their diabetes than those who had not completed any education. Given the complexities surrounding the association between individual-level socioeconomic status and health in LMIC undergoing nutritional and epidemiologic transitions, future research should continue to investigate these associations.

Employment status and education level were not significantly associated with any of the remaining diabetes outcomes. Importantly, our outcomes were all nested, meaning that the analytic sample decreased throughout the progression of the diabetes care cascade. Compared to the significant associations we observed between employment status, educational attainment, and diabetes prevalence (n = 3,685), our null findings for diabetes awareness (n = 830) and control (n = 629) are likely due, at least in part, to a loss of statistical power to detect significant effect sizes. To rule this out, future research should examine the individual-level socioeconomic determinants of these care cascade outcomes in larger Black South African samples.

Our analyses revealed an inverse association between diabetes prevalence and the most recent estimates of relative and absolute area-level deprivation, which is consistent with existing literature in LMICs [46]. We observed that Black South Africans living in more deprived wards had lower diabetes prevalence and a lower prevalence of a previous diabetes diagnosis. Whether the latter association is driven by a lower overall prevalence of diabetes in these communities versus reduced access to healthcare institutions remains an important point of discussion. Existing research has indicated that more deprived regions in South Africa have seen increases in screening for diabetes in recent years, lending support to the former explanation [47]. Moreover, our analysis found that increases in absolute deprivation from 2001 to 2011 were associated with an *increased* prevalence of diabetes at the time of the study. Keeping existing work in mind, areas that have become more deprived over time may have received

increased attention from public health and medical institutions, NGOs, and researchers to improve screening and diabetes diagnosis, resulting in a reported higher prevalence of diabetes in these communities [47–49].

The associations between (absolute and relative) changes in area-level deprivation and diabetes control were robust and deserve commentary. We found that Black South Africans living in wards that became more deprived, both in terms of absolute and relative deprivation, from 2001 to 2011 had a higher prevalence of diabetic control, yet no difference in terms of their access to treatment. A common explanation for this association is that it is driven by the changes in lifestyle that typically accompany urbanization, including increased access to obesogenic foods and sedentary lifestyles [50,51]. Indeed, South Africans living in rural, more deprived settings generally do not have the means to afford obesogenic foods, and in some cases rely on gardening and farming for sustenance [52–55]. Furthermore, the geographic isolation of these communities may result in increased physical activity, as residents may be required to walk long distances to shop, socialize, or access healthcare institutions.

However conceivable it may be that lifestyle and nutritional choices may be driving our observed associations, it should be recognized that increased physical activity and nutritious food options are not universal for Black South Africans living in deprived communities, and those with increasing deprivation over time. Several studies have indicated that affordable unhealthy foods are becoming more prevalent in rural, more deprived, areas in South Africa as large fast-food chains expand in these regions [56–60]. Given these inconsistencies in the literature, perhaps a more robust question to ask is: what is driving the worse health of Black South Africans who are living in areas that may be slightly more prosperous than the most severely deprived areas?

It is plausible that Black South Africans living in less deprived wards experience more of the social stressors that accompany living in more developed areas, which are primarily occupied by White South Africans; including interpersonal and structural racism. It should also be acknowledged that although on average, these wards may be less deprived at the area level, it is likely that the Black Africans living in these areas are segregated to the least desirable and most unsafe living conditions, leading to disproportionate exposure to environmental pollutants, overcrowding, and crime [61–64]. Although steady progress has been made to de-segregate South Africa since the fall of Apartheid, vast neighborhood-level socioeconomic inequities persist [65–70]. Research from the United States, another country with a long history of institutional racism and marked racial segregation, has been investigating for some time the contemporary health impacts of racial segregation that was enacted into policy under the Home Owners' Loan Act in 1933. Research in Seattle, Washington, for example, has demonstrated that individuals currently living in areas that experienced historical racial segregation through the process of "redlining," experienced higher rates of diabetes mortality and years of life lost compared to those residing in less segregated areas [71]. It has been hypothesized that individuals who live in these areas of intergenerational harm are more likely to be exposed to various social stressors, including discrimination, mass incarceration, housing instability, and labor market exclusion, all social determinants of diabetes (and more broadly health) [72]. The impact of historical redlining on contemporary health outcomes has been explored across several conditions, including cardiovascular disease, cancer, and preterm birth [73–75]. While some existing research in South Africa has examined the current impact of analogous policies on tuberculosis susceptibility, scant attention has been given to chronic diseases [76]. As South Africa, like the United States, continues to reckon with its history of racial segregation and trauma, the scientific community should continue to investigate the contemporary consequences of those policies and strive to identify factors that lessen their harm.

Our work, while valuable in identifying the socioeconomic correlates of the diabetes care cascade outcomes among Black South Africans, is not without its limitations. Although our study has a comparatively large sample size and collected data from several districts, the sample was purposively selected, thus selection bias and unmeasured confounding may have impacted our reported associations. Additionally, our sample was restricted to Black South Africans aged 45 and older, residing in wards that were on average more deprived relative to the rest of South Africa. While this may limit the generalizability of our findings, 45 is a clinically relevant age for diabetes development, and Black South Africans are more likely to live in deprived areas than members of other racial groups [77]. While our findings for area-level deprivation were robust, it must be recognized that, although we used the most recently available estimates, they come from 2011, and this may not reflect the area-level deprivation of the wards when the study was collected in 2017/2018. Finally, we lack information about the temporal ordering of our variables, and as such, we cannot comment on the causal nature of our observed relationships.

Despite its limitations, our work provides vital information about the socioeconomic distribution of diabetes outcomes in Black South African communities and has implications for public health interventions and policies. It suggests that Black South Africans living in less deprived wards have a higher prevalence of diabetes, and those that have not experienced drastic changes in deprivation have a lower prevalence of diabetic control. The evidence of educational initiatives to improve diabetes outcomes has had variable results. One hospital-based randomized controlled trial in South Africa that did not demonstrate significant reductions in HbA1c indicated that barriers to program success included a lack of trained professionals and well-resourced settings for program activities to occur in, resulting in high rates of study dropout [78]. Another trial in a resource-limited setting indicated that despite increased reported knowledge of diabetes and feelings of autonomy among program attendees, the authors did not observe a significant reduction in HbA1c [79]. At the same time, community-based interventions have shown promise in improving diabetes detection and management. For example, a 2011 study found that participants of a program that conducted monthly educational diabetes clinics using a patient empowerment model had significant reductions in their HbA1c after four years [80]. Similar benefits have been achieved in other community-based interventions, especially among those utilizing community health workers to link individuals to local healthcare facilities [32,81]. Future research should examine the feasibility of these community-based programs in areas that are more economically developed.

At the policy level, there is wide-ranging consensus within the scientific community that taxation on sugar-sweetened beverages is a cost-effective intervention to reduce diabetes mortality, disability-adjusted life years, and obesity [82–85]. Furthermore, existing analyses have projected that the benefits of this policy would be experienced equitably–South Africans in the lowest income bracket (and disproportionately Black South Africans) were expected to experience the strongest decrease in out-of-pocket medical payments [86]. In 2018, following this evidence, South Africa imposed a 10% tax on sugar-sweetened beverages, and existing observational research has indicated that this led to a reduction in the consumption of taxed beverages, especially among low-income houses, with the authors suggesting that sugar-based taxation is an effective public health prevention measure [87].

Finally, and most notably, future research should continue to investigate the pathways by which Apartheid's legacy of racial segregation may be continuing to impact contemporary racial segregation (and its resultant detrimental economic and health impacts), and how this historical trauma may be embodied in the lived experience of Black South Africans.

## Supporting information

**S1 File. Healthrise diabetes dataset noPII.** Anonymized HealthRise Dataset.
(DTA)

**S2 File. Inclusivity questionnaire diabetes.** Inclusivity in global research questionnaire.
(DOCX)

## Author Contributions

**Conceptualization:** Sanele Listen Mandlenkosi Madela, Anthony David Mbewu, David R Williams, Sibusiso Sifunda, Anam Nyembezi, Sasiragha Priscilla Reddy.

**Data curation:** Nigel Walsh Harriman, Ronel Sewpaul.

**Formal analysis:** Nigel Walsh Harriman, Ronel Sewpaul.

**Funding acquisition:** Sanele Listen Mandlenkosi Madela.

**Investigation:** Anam Nyembezi, Sasiragha Priscilla Reddy.

**Methodology:** Nigel Walsh Harriman, Ronel Sewpaul.

**Writing – original draft:** Sanele Listen Mandlenkosi Madela, Nigel Walsh Harriman, David R Williams, Sasiragha Priscilla Reddy.

**Writing – review & editing:** Nigel Walsh Harriman, Ronel Sewpaul, Anthony David Mbewu, David R Williams, Sibusiso Sifunda, Thabang Manyaapelo, Anam Nyembezi, Sasiragha Priscilla Reddy.

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
