## [Decision Letter · Decision Letter 0]

19 Apr 2023

PONE-D-22-34990Area-Level Deprivation and Individual-Level Socioeconomic Correlates of the Diabetes Care Cascade Among Black South Africans in uMgungundlovu, KwaZulu-Natal, South AfricaPLOS ONE

Dear Dr. Harriman,

Thank you for submitting your manuscript to PLOS ONE. After careful consideration, we feel that it has merit but does not fully meet PLOS ONE’s publication criteria as it currently stands. Therefore, we invite you to submit a revised version of the manuscript that addresses the points raised during the review process.

As you can see from the full reports included below, your manuscript has been evaluated by two reviewers. They appreciate the relevance and importance of your research question, but also raised several concerns about the methodology, statistical analysis, and the interpretation of the results. Please carefully address all points raised. 

We look forward to receiving your revised manuscript.

Kind regards,

Dario Ummarino, PhD

Senior Editor

PLOS ONE

Additional Editor Comments:

Authors are using data from a cross-sectional study but opted to use logistic regression. Given that the study outcome (diabetes) they reported is common this population (23%), the use of logistic regression and Odds ratio is not appropriate. Alternative to logistic regression models must be used and prevalence ratios must be estimated instead.

Reviewers' comments:

Reviewer's Responses to Questions

**Comments to the Author**

1. Is the manuscript technically sound, and do the data support the conclusions?

Reviewer #1: Partly

Reviewer #2: Yes

2. Has the statistical analysis been performed appropriately and rigorously? 

Reviewer #1: No

Reviewer #2: Yes

3. Have the authors made all data underlying the findings in their manuscript fully available?

Reviewer #1: Yes

Reviewer #2: Yes

4. Is the manuscript presented in an intelligible fashion and written in standard English?

Reviewer #1: Yes

Reviewer #2: Yes

5. Review Comments to the Author

Reviewer #1: Area-Level Deprivation and Individual-Level Socioeconomic Correlates of the Diabetes Care Cascade Among Black South Africans in uMgungundlovu, KwaZulu-Natal, South Africa

Comments

The manuscript presents the results of a study exploring the association between individual and area-level socioeconomic variables and diabetes prevalence, awareness, treatment, and control within a sample of Black South Africans aged 45 years or older. Employment status and educational attainment were considered as socioeconomic indicators at the individual level, and the Multidimensional Poverty Index (SAMPI) score and its changes between 2001 and 2011 as socioeconomic indicators at the area (ward) level.

The authors performed a secondary analysis of baseline data collected on 3685 participants in a community-based program designed to improve screening, diagnosis, management, and control of hypertension and diabetes in underserved communities.

Logistic regression was applied to assess the relationships between four binary outcomes (diabetes diagnosis, awareness, treatment, control) and a series of predictors, adjusting for age, sex, body mass index category and hypertension diagnosis as potential confounders. Multiple models were fitted, with different sets of predictors. Multiple imputation was used to deal with missing data in the variables of interest.

The objectives of the study are of interest and the finding can constitute a useful contribution to understanding the complex relationships between socioeconomic variables and diabetes prevalence, awareness, treatment and control and their implications in terms of policies and prioritization of preventive interventions. The Authors should be commended for this.

However, in my opinion, some methodological aspects of the study need further attention from the authors in terms of clarification and/or integration. My specific comments/suggestions are listed below.

Major comments

1) Interpretation of changes in SAMPI quintile

The Authors used changes in SAMPI quintiles (as opposed to the scores themselves) as a measure of change in area-level deprivation. However, SAMPI quintiles are a relative measure of deprivation, which compares deprivation in a ward with all other wards in the Country. Therefore, the fact that the deprivation quintile of a ward has changed between 2001 and 2011 says nothing about changes in the poverty headcount/poverty intensity in the ward: it is well possible that the change in quintile of a ward is due only to changes in the distribution of the SAMPI in the remaining wards (which defines the cut-off for the quintiles). This affects the interpretation of the nominal indicator included in the models, which cannot per-se be considered an indicator of an actual change of poverty/deprivation in the ward.

The Authors might want to consider using an indicator of change constructed using the actual SAMPI score rather than its quintile and/or clarify the correct interpretation of the changes in the quintile.

2) Statistical models

The Authors modelled each of the four outcomes using logistic regression, where the relevant predictors were added in blocks. This approach is described a “hierarchical logistic regression”. However, the hierarchical nature of the modelling procedure is ignored, and the possibility of statistically comparing the fit of the models (made possible by the nested nature of the models) is not exploited at all, and no comparable measures of fit (such a likelihood and/or information indices) are reported for the various models. It would have been interesting, especially with regard to the analysis of the explanatory power of the area-level variables, to assess if the introduction of additional variables improved significantly the fit of the model.

Also, while the choice of introducing are-level variables at individual level is a legitimate one in the light of model simplicity, a suggestion that the Authors might want to consider is the use of a multilevel model (with individual predictors/confounders at the first level, and area-level variables at the second level) and a way to better understand the relative contribution of these different types of variables in explaining the observed variance of the outcomes.

3) SAMPI scores time reference

The SAMPI scores used as area-level socioeconomic indicators refer to the year 2011, while data collection was carried out in 2017/18. The author should acknowledge this limitation and discuss the potential implications, especially when discussing changes in deprivation.

Minor comments

1) Definition of control (Page 8, line 161)

The cut-off to define diabetes control seems to me quite high. A post-prandial cut-off of 10 mmol/L seems to be more common.

2) SAMPI quintiles (page 9, line 192)

“SAMPI scores were modeled as a nominal variable”: more precisely, an ordinal variable.

3) Historical changes in area-level deprivation (page 14, line 316)

See comments above regarding the interpretability of the chosen indicator as “historical change in deprivation” and the time frame of the assessed changes in the SAMPI compared to the data collection period.

4) Interpretation of the relationship involving employment status (page 15 and other)

Given the age range considered, it is possible that many respondents were unemployed because they had reached retirement age. This might have diluted the relationships of interests.

Reviewer #2: 1. How was the sample size calculated? Why were 3685 Balck South Africans selected and how?

2. The definitions of dependent variables are sound.

3. It may be explained why no international guideline for cut-off of diabetes was used!

6. PLOS authors have the option to publish the peer review history of their article (what does this mean?). If published, this will include your full peer review and any attached files.

Reviewer #1: No

Reviewer #2: No

---

## [Author Response · Author response to Decision Letter 0]

26 Jun 2023

Please note that the editor has requested that we re-run the analysis and present prevalence ratios. As such, we have incorporated both your and their feedback regarding the regression models.

Reviewer #1: 

Area-Level Deprivation and Individual-Level Socioeconomic Correlates of the Diabetes Care Cascade Among Black South Africans in uMgungundlovu, KwaZulu-Natal, South Africa

Comments

The manuscript presents the results of a study exploring the association between individual and area-level socioeconomic variables and diabetes prevalence, awareness, treatment, and control within a sample of Black South Africans aged 45 years or older. Employment status and educational attainment were considered as socioeconomic indicators at the individual level, and the Multidimensional Poverty Index (SAMPI) score and its changes between 2001 and 2011 as socioeconomic indicators at the area (ward) level.

The authors performed a secondary analysis of baseline data collected on 3685 participants in a community-based program designed to improve screening, diagnosis, management, and control of hypertension and diabetes in underserved communities.

Logistic regression was applied to assess the relationships between four binary outcomes (diabetes diagnosis, awareness, treatment, control) and a series of predictors, adjusting for age, sex, body mass index category and hypertension diagnosis as potential confounders. Multiple models were fitted, with different sets of predictors. Multiple imputation was used to deal with missing data in the variables of interest.

The objectives of the study are of interest and the finding can constitute a useful contribution to understanding the complex relationships between socioeconomic variables and diabetes prevalence, awareness, treatment and control and their implications in terms of policies and prioritization of preventive interventions. The Authors should be commended for this.

However, in my opinion, some methodological aspects of the study need further attention from the authors in terms of clarification and/or integration. My specific comments/suggestions are listed below.

Major comments

1) Interpretation of changes in SAMPI quintile

The Authors used changes in SAMPI quintiles (as opposed to the scores themselves) as a measure of change in area-level deprivation. However, SAMPI quintiles are a relative measure of deprivation, which compares deprivation in a ward with all other wards in the Country. Therefore, the fact that the deprivation quintile of a ward has changed between 2001 and 2011 says nothing about changes in the poverty headcount/poverty intensity in the ward: it is well possible that the change in quintile of a ward is due only to changes in the distribution of the SAMPI in the remaining wards (which defines the cut-off for the quintiles). This affects the interpretation of the nominal indicator included in the models, which cannot per-se be considered an indicator of an actual change of poverty/deprivation in the ward.

The Authors might want to consider using an indicator of change constructed using the actual SAMPI score rather than its quintile and/or clarify the correct interpretation of the changes in the quintile.

We sincerely thank the reviewer for this comment – it is well-taken and greatly appreciated. Moreover, this is also relevant to our measure of deprivation in 2011. As such, we have added two absolute area-level measures to this analysis. 

1. Absolute deprivation in 2011, reflecting a continuous measure of log-transformed SAMPI 2011 scores. Given the strong skew in SAMPI scores, we transformed these scores to ensure our variable was linear to (the natural log of) our outcome. To give you a sense of the skew, the highest score for all of Africa in 2011 was approximately 0.228 (Range 0-1), and the mean score was 0.04. 

2. Absolute change in deprivation from 2001 to 2011 using the actual SAMPI score. This score was constructed by subtracting the 2001 score from the 2011 score (positive values indicate worsening deprivation). In our analyses, to enhance the interpretability, we multiplied this difference by 100 so that we could infer the association between our outcomes and a 0.01-unit difference in change in SAMPI score. Without this second step, a 1-unit change becomes challenging to interpret, given that SAMPI scores range from 0-1.

Again, we thank the reviewer for their wise suggestion, we believe this makes our findings much more methodologically robust and of greater use to policymakers, researchers, and practitioners.

2) Statistical models

The Authors modelled each of the four outcomes using logistic regression, where the relevant predictors were added in blocks. This approach is described a “hierarchical logistic regression”. However, the hierarchical nature of the modelling procedure is ignored, and the possibility of statistically comparing the fit of the models (made possible by the nested nature of the models) is not exploited at all, and no comparable measures of fit (such a likelihood and/or information indices) are reported for the various models. It would have been interesting, especially with regard to the analysis of the explanatory power of the area-level variables, to assess if the introduction of additional variables improved significantly the fit of the model.

Thank you for this suggestion. While we agree with the reviewer that this is a noteworthy investigation, our analysis does not aim to generate a predictive model of diabetes (in which case, we would absolutely pursue this analysis), yet rather it describes the associations between individual and area-level SES and diabetes outcomes. We wholeheartedly agree with the reviewer’s feedback as to how we might improve our paper methodologically to meet that end and have re-ran all analyses using new area-level variables (see above) and a multi-level model (see below). Given that the changes that the reviewer has requested both better focus the analysis on its goals, and make it more methodologically robust, we prefer to keep those findings at the center of the paper, rather than to shift the focus onto model selection. Please let us know if you do not agree, and again, thank you for your methodological improvements.

Also, while the choice of introducing are-level variables at individual level is a legitimate one in the light of model simplicity, a suggestion that the Authors might want to consider is the use of a multilevel model (with individual predictors/confounders at the first level, and area-level variables at the second level) and a way to better understand the relative contribution of these different types of variables in explaining the observed variance of the outcomes.

We greatly appreciate this recommendation and wholeheartedly agree with the reviewer that a multilevel model should be used. As such, we have re-run all analyses using a multilevel model with a randomly varying intercept for ward. Again, we thank the reviewer for their suggestion as to how we might make our findings more methodologically robust.

3) SAMPI scores time reference

The SAMPI scores used as area-level socioeconomic indicators refer to the year 2011, while data collection was carried out in 2017/18. The author should acknowledge this limitation and discuss the potential implications, especially when discussing changes in deprivation.

We agree with the reviewer and have added a sentence to the discussion to reflect this (line 476-479).

Minor comments

1) Definition of control (Page 8, line 161)

The cut-off to define diabetes control seems to me quite high. A post-prandial cut-off of 10 mmol/L seems to be more common.

Thank you, we have made this change and re-ran the analyses to reflect this.

2) SAMPI quintiles (page 9, line 192)

“SAMPI scores were modeled as a nominal variable”: more precisely, an ordinal variable.

Agreed, thank you for your attention to detail. 

3) Historical changes in area-level deprivation (page 14, line 316)

See comments above regarding the interpretability of the chosen indicator as “historical change in deprivation” and the time frame of the assessed changes in the SAMPI compared to the data collection period.

We appreciate this comment, please let us know if you still have concerns given the updated analyses.

4) Interpretation of the relationship involving employment status (page 15 and other)

Given the age range considered, it is possible that many respondents were unemployed because they had reached retirement age. This might have diluted the relationships of interests.

We agree with this point and have added a sentence to the discussion to reflect this (line 376-379).

 

Reviewer #2: 

Please note that the editor has requested that we present prevalence ratios, and as per the author reviewer’s requests, we have added two independent area-level variables, and fit a multi-level model.

1. How was the sample size calculated? Why were 3685 Black South Africans selected and how?

We have added additional sampling criteria to the methods (143-146), however, the latter question is already answered in the methods under the statistical analyses section (214-218).

2. The definitions of dependent variables are sound.

Thank you! 

3. It may be explained why no international guideline for cut-off of diabetes was used!

Thank you for this comment, we have revised the analysis considerably, including changing the cutoff for diabetes to 11.1 mmol/l, and the cutoff for diabetic control to 10.0 mmol/l, as per the International Diabetes Foundation guidelines.

---

## [Decision Letter · Decision Letter 1]

10 Oct 2023

Area-Level Deprivation and Individual-Level Socioeconomic Correlates of the Diabetes Care Cascade Among Black South Africans in uMgungundlovu, KwaZulu-Natal, South Africa

PONE-D-22-34990R1

Dear Dr. Harriman,

We’re pleased to inform you that your manuscript has been judged scientifically suitable for publication and will be formally accepted for publication once it meets all outstanding technical requirements.

Kind regards,

Ryan G Wagner, MSc(Med), MBBCh, PhD

Academic Editor

PLOS ONE

Additional Editor Comments (optional):

Reviewers' comments:

Reviewer's Responses to Questions

**Comments to the Author**

1. If the authors have adequately addressed your comments raised in a previous round of review and you feel that this manuscript is now acceptable for publication, you may indicate that here to bypass the “Comments to the Author” section, enter your conflict of interest statement in the “Confidential to Editor” section, and submit your "Accept" recommendation.

Reviewer #3: All comments have been addressed

2. Is the manuscript technically sound, and do the data support the conclusions?

Reviewer #3: Yes

3. Has the statistical analysis been performed appropriately and rigorously? 

Reviewer #3: Yes

4. Have the authors made all data underlying the findings in their manuscript fully available?

Reviewer #3: Yes

5. Is the manuscript presented in an intelligible fashion and written in standard English?

Reviewer #3: Yes

6. Review Comments to the Author

Reviewer #3: Interpreting SAMPI data is critical to how policy-makers respond to the evidence. The suggested corrections by Reviewer 1 are extremely important, including the cut off scores used to define diabetes. These have been adequately addressed

7. PLOS authors have the option to publish the peer review history of their article (what does this mean?). If published, this will include your full peer review and any attached files.

Reviewer #3: **Yes: **Arvin Bhana

---

## [Editor Report · Acceptance letter]

30 Nov 2023

PONE-D-22-34990R1 

Area-Level Deprivation and Individual-Level Socioeconomic Correlates of the Diabetes Care Cascade Among Black South Africans in uMgungundlovu, KwaZulu-Natal, South Africa 

Dear Dr. Harriman:

I'm pleased to inform you that your manuscript has been deemed suitable for publication in PLOS ONE. Congratulations! Your manuscript is now with our production department. 

Kind regards, 

on behalf of

Dr. Ryan G Wagner 

Academic Editor

PLOS ONE